# Prevalence of low birth weight and its associated factor at birth in Sub-Saharan Africa: A generalized linear mixed model

**Zemenu Tadesse Tessema** *, **Koku Sisay Tamirat, Achamyeleh Birhanu Teshale, Getayeneh Antehunegn Tesema**

Department of Epidemiology and Biostatistics, Institute of Public Health, College of Medicine and Health Sciences, University of Gondar, Gondar, Ethiopia

* zemenut1979@gmail.com

## Abstract

### Background

Low birth weight (LBW) is one of the major determinants of perinatal survival, infant morbidity, and mortality, as well as the risk of developmental disabilities and illnesses in future lives. Though studies were conducted to assess the magnitude and associated factors of low birth weight, most of the studies were at a single center and little information on the regional level. Hence, this study assessed the prevalence and associated factors of low birth weight in Sub-Saharan countries.

### Method

This study was based on secondary data sources from 35 Sub-Saharan countries' Demography and Health Survey (DHS). For this study, we used the Kids Record (KR file) data set. In the KR file, all under-five children who were born in the last five years preceding the survey in the selected enumeration area who had birth weight data were included for the study. To identify determinants of low birth weight multivariable mixed-effect logistic regression model fitted. Adjusted Odds Ratios (AOR) with a 95% Confidence Interval (CI) and p-value ≤0.05 in the multivariable model were used to declare significant factors associated with low birth weight at birth.

### Result

The pooled prevalence of newborn babies' low birth weight measured at birth in Sub-Saharan Africa was 9.76% with (95% CI: 9.63% to 9.89%). Female child, women not participated in healthcare decision making, and wider birth intervals, divorced/ separated women, and twin pregnancies associated with increased occurrences of low birth weight, while some level of woman and husband education, antenatal care visits, older maternal age, and multiparity associated with reduced occurrence low birth weight.

**Data Availability Statement:** All relevant data are within the manuscript.

**Funding:** We didn't receive external funds for this research.

**Competing interests:** The authors declared that they have no competing interests.

**Abbreviations:** ANC, Antenatal Care; AOR, Adjusted Odds Ratio; Area, Under the Curve; CI, Confidence Interval; DHS, Demographic and Health Survey; EA, Enumeration Area; LBW, Low Birth Weight; LLR, Likelihood Ratio; NPV, Negative Predictive Value; PPV, Positive Predictive Value; ROC, Reciever Operating Curve; SD, Standard Deviation; SGD, Sustainable Development Goal; SSA, Sub-Saharan Africa; USA, United States of America.

## Conclusion

This study revealed that the magnitude of low birth weight was high in sub-Saharan Africa countries. Therefore, the finding suggests that more emphasis is important for women with a lack of support, multiples, and healthcare decision-making problems.

## Introduction

Low birth weight(LBW) is one of the major determinants of perinatal survival, infant morbidity, mortality, and the risk of developmental disabilities and illnesses in future lives [1,2]. Globally, WHO estimates that about 30 million low birth weight babies are born annually (23.4% of all births), and they often face short and long-term health consequences [3,4]. Half of all low birth weight babies are born in South-central Asia, where 27 percent are below 2500g at birth, while LBW levels in sub-Saharan Africa are estimated at 15 percent [4,5].

LBW is determined by two major processes, which are the duration of gestation and intrauterine growth rate. Several studies indicated that a baby's low weight at birth is either the result of preterm birth (before 37 weeks of gestation) or restricted fetal (intrauterine) growth [5–8]. There is a strong consensus that birth weight plays an important role in infant mortality, morbidity, development, and future health of the child [9]. Particularly, low birth weight is the most significant risk factor for adverse health outcomes, including common childhood diseases [9]. The association between LBW and a greatly elevated risk of infant mortality and other physical and neurological impairments are well-established facts [10]. The risk of neonatal mortality for very LBW infants is 25 to 30 times greater than for infants with a birth weight exceeding 2500g, and it increases sharply as birth weight decreases [10].

Demographic risk factors include young maternal age, primiparity, and low education level, and poor maternal nutritional status–both before and during pregnancy—which are well-recognized determinants of birth outcomes [2,5–7,10–13]. Empirical studies from developed and developing countries show that maternal undernutrition, chronic illnesses like HIV/AIDS, and cardiovascular diseases are associated with low birth weight [14,15]. Focused antenatal care, nutritional counseling during the prenatal and perinatal period, and institutional deliveries are some of the interventions to avert low birth weight. To achieve the targets of reducing child mortality of the Sustainable Development Goals (SDGs)-2030, it is necessary to generate adequate evidence on the magnitude of low birth weight and associated factors to contribute to the development of timely interventions. Studies were conducted on LBW at birth among mothers who gave birth in different countries [4,16]. However, the results of the studies were inconsistent; factors that had an association in some studies may be had no association in other studies and vice-versa. Moreover, to the best of our knowledge, there is no pooled data on LBW magnitudes and associate factors at the Sub-Saharan African level.

Hence, this study assessed the pooled prevalence and associated factors of low birth weight in Sub-Saharan countries. This study could give additional knowledge about the risk factors of low birth weight in low-income countries and for evidence-based interventions. Moreover, this study's result could support policymakers, clinicians, and programmers to design interventions on preventing LBW.

## Method

### Data source

Secondary data analysis was done based on the most recent Demographic and Health Surveys (DHS) conducted in the 35 Sub-African (SSA) countries. Southern Region of Africa (Lesotho,

Namibia and South Africa), Central Region of Africa(Angola, DR Congo, Congo, Cameroon, Gabon, Sao Tome & Principe, and Chad), Eastern Region of Arica (Burundi, Ethiopia, Kenya, Comoros, Madagascar, Malawi, Mozambique, Rwanda, Tanzania, Uganda, Zambia, and Zimbabwe), Western Region of Africa (Burkina-Faso, Benin, Cote d'Ivoire, Ghana, Gambia, Guinea, Liberia, Mali, Nigeria, Niger, Sierra Leone, Senegal, and Togo). Each country's sampling procedure was the same or homogeneity across countries [17] (**Table 1**). These datasets were appended together to investigate mothers' perceived birth size for the prediction of low birth weight babies in SSA. The DHS is a nationally representative survey that collects data on basic health indicators like mortality, morbidity, family planning service utilization, fertility, maternal and child health. The data were derived from the measure DHS program. The DHS has different datasets (men, women, children, birth, and household datasets). For this study,

**Table 1. The study country, DHS year and number of study participants in SSA.**

| Country | DHS year | Low Birth weight | Sample size (202,878) |
|---|---|---|---|
| Angola | 2014 | 778 | 7326 |
| Benin | 2013 | 985 | 13882 |
| Burkina Faso | 2016 | 740 | 7372 |
| Burundi | 2015/16 | 527 | 6897 |
| Cameron | 2013/14 | 647 | 4585 |
| Chad | 2011/12 | 123 | 1484 |
| Comoros | 2011 | 157 | 2221 |
| Congo | 2012 | 1092 | 10921 |
| Cote de vaire | 2008/09 | 198 | 1499 |
| Dr Congo | 2014/15 | 461 | 6123 |
| Ethiopia | 2010 | 349 | 2145 |
| Gabon | 2016 | 643 | 5036 |
| Gambia | 2014 | 1789 | 14557 |
| Ghana | 2012 | 841 | 5980 |
| Guinea | 2008/09 | 465 | 7376 |
| Kenya | 2015/16 | 439 | 6369 |
| Lesotho | 2011 | 985 | 10254 |
| Liberia | 2014/15 | 672 | 7882 |
| Madagascar | 2015/16 | 498 | 5268 |
| Malawi | 2011 | 363 | 2517 |
| Mali | 2018 | 531 | 4071 |
| Mozambique | 2013/14 | 174 | 2268 |
| Namibia | 2010 | 450 | 3123 |
| Niger | 2017 | 1351 | 9765 |
| Nigeria | 2011 | 949 | 8182 |
| Rwanda | 2014 | 634 | 4460 |
| Saotome and Principe | 2013 | 321 | 3416 |
| Senegal | 2018 | 550 | 4685 |
| Sierra Leone | 2013 | 396 | 3841 |
| South Africa | 2018 | 146 | 1509 |
| Tanzania | 2018 | 589 | 8048 |
| Togo | 2012 | 365 | 3082 |
| Uganda | 2010/11 | 410 | 5786 |
| Zambia | 2010/11 | 1092 | 6936 |
| Zimbabwe | 2013/14 | 379 | 3997 |

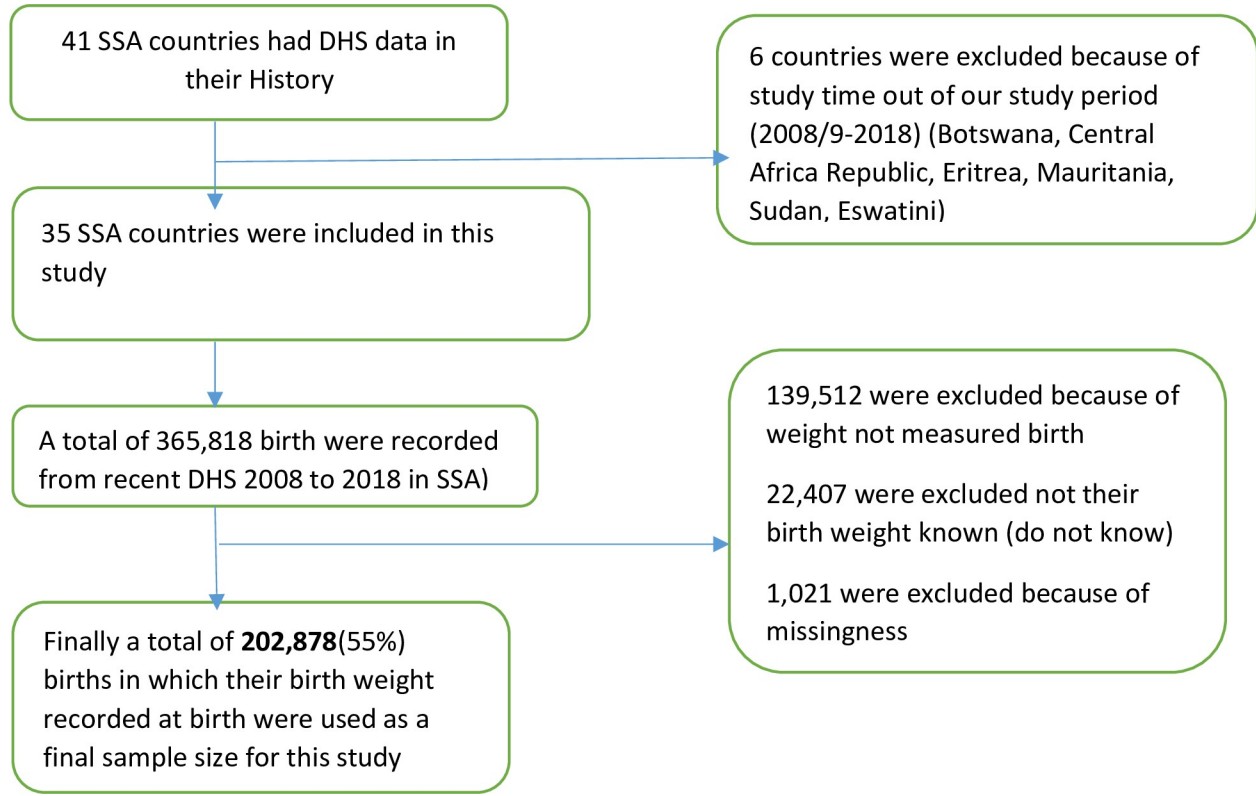

**Fig 1. Flow chart of data for low birth weight in SSA countries using recent Demographic and Health Surveys.**

we used the Kids Record (KR file) data set. In the KR file, all under-five children who were born in the last five years preceding the survey in the selected enumeration area who had birth weight data were included for the study. The DHS used two stages of stratified sampling technique to select the study participants. We pooled the DHS surveys conducted in the 35 Sub-African countries, and a total weighted sample of 202,878 under-five children was included in the study (**Fig 1**).

## Variables of the study

**Dependent variable.** The main outcome variable of this study was birth weight. Data on children's birth weight were collected from mothers who gave birth within five years before the survey of each Sub-Saharan Africa country either by accessing birth weight through record review or by the mother's report by recalling the measured weight of the child at birth. The births without recorded birth weight were excluded from the study. Finally, LWB was defined as a birth weight <2.5kg, and those 2.5 kgs were considered normal and above normal birthweights.

**Independent variables.** Potential risk factors for LBW were included based on the literature review [4,11,12,16,18,19], we included two types of variables in the analysis. Level one variable (individual-level variables) that is maternal and husband education (has no education, primary education and secondary and above, no education means respondents cannot read and write or had no any formal education), maternal age, mother marital status, household wealth index, maternal occupation status, women health care decision making autonomy, media exposure, number of antenatal care (ANC) visit, preceding birth interval, parity

(recoded as 1–2,3–5 and 6+), sex of the child, type of birth and iron supplementation. Level two variables (community-level variables) included in this study were region (recoded as West Africa, East Africa, Central Africa, and South Africa), residence, and country.

## Operational definition

**Household wealth index.** The wealth variable was generated from the wealth index for the households. In the dataset, the index has five quintiles, such as; the lowest quintile (poorest), second quintile (poorer), third quintile (middle), four quintiles (wealthier), and the fifth quintile (wealthiest). In this study for ease of analysis, this variable was categorized as 'poorest' and 'poorer' were coded as (1) 'poor,' the middle was coded as (2) 'middle,' and 'wealthier' and 'wealthiest' were coded as (3) 'rich".

**Women health care decision-making autonomy.** Defines as women health care decision-making capacity for a woman to achieve well-being and decision making a role.

**Media exposure.** A respondent said to be media exposed if they listen/read at least one media in the week (Radio or TV or Newspapers)

## Data management and analysis

We pooled the data from the 35 Sub-Saharan African countries together after extracting the variables based on literature. Before any statistical analysis, the data were weighted using sampling weight, primary sampling unit, and strata to restore the survey's representativeness and take sampling design when calculating standard errors and reliable estimates. Cross tabulations and summary statistics were done using STATA version 14 software. A meta-analysis was done using the "meta-prop" Stata command. A fixed-effect meta-analysis was done to estimate the pooled prevalence of LBW in SSA. Pooled analysis was done for both SSA regions and sub-level regions. The pooled prevalence of low birth weight at birth with the 95% Confidence Interval (CI) was reported using a forest plot. For the determinants factors, the DHS data had a hierarchical structure; this violates the independence of observations and equal variance assumption of the traditional logistic regression model. Hence, children are nested within a cluster, and we expect that children within the same cluster may be more similar to each other than women in the rest of the country. This implies that there is a need to take into account the between cluster variability by using advanced models. Therefore, a mixed effect logistic regression model (both fixed and random effect) was fitted. Since the outcome variable was binary, standard logistic regression and Generalized Linear Mixed Models (GLMM) were fitted. Model comparison and fitness were made based on the Intra-class Correlation Coefficient (ICC), Likelihood Ratio (LR) test, Median Odds Ratio (MOR), and deviance (-2LLR) values since the models were nested. The model with the lowest deviance was chosen. Accordingly, the mixed-effect logistic regression model was the best-fitted model. Variables with a p-value <0.2 in the bi-variable analysis were considered in the multivariable mixed-effect logistic regression model. Adjusted Odds Ratios (AOR) with a 95% Confidence Interval (CI) and p-value ≤0.05 in the multivariable model were used to declare significant factors associated with low birth weight at birth.

## Ethical consideration

Permission for data access was obtained from a major demographic and health survey through an online request from http://www.dhsprogram.com. The data used for this study were publicly available with no personal identifier.

# Result

## Socio-demographic and economic characteristics

A total of 202,878 mothers were included; of these, 83,512 (41.0%) were in West Africa. The majority (58.3%) of the mothers were rural residents, and 103,763 (51.0%) were aged 20–29 years. About 76,879 (37.8%) of mothers had attained primary level of education, and 52.7% of their husbands attained secondary education and above (**Table 3**).

## The pooled prevalence of low birth weight measured at birth in Sub-Saharan Africa

The pooled prevalence of newborn babies' low birth weight measured at birth in Sub-Saharan Africa was 9.76%, with a 95% confidence interval from 9.63% to 9.89%. The highest LBW was recorded in Ethiopia, 16.21%, with its 95% confidence interval of 14.71 to 17.83. The lowest LBW was recorded in Guinea, 6.30%, with a 95% confidence interval of 5.75 to 6.85 (**Fig 2**).

## Model comparison

Log-likelihood and deviance were checked, and the mixed effect logistic regression model was chosen because of the smallest value of deviance since the model (Table 2). Furthermore, MOR 1.08 indicates if we randomly select two women from different enumeration areas, women from enumeration areas with low birth weight were increased by 8% than women

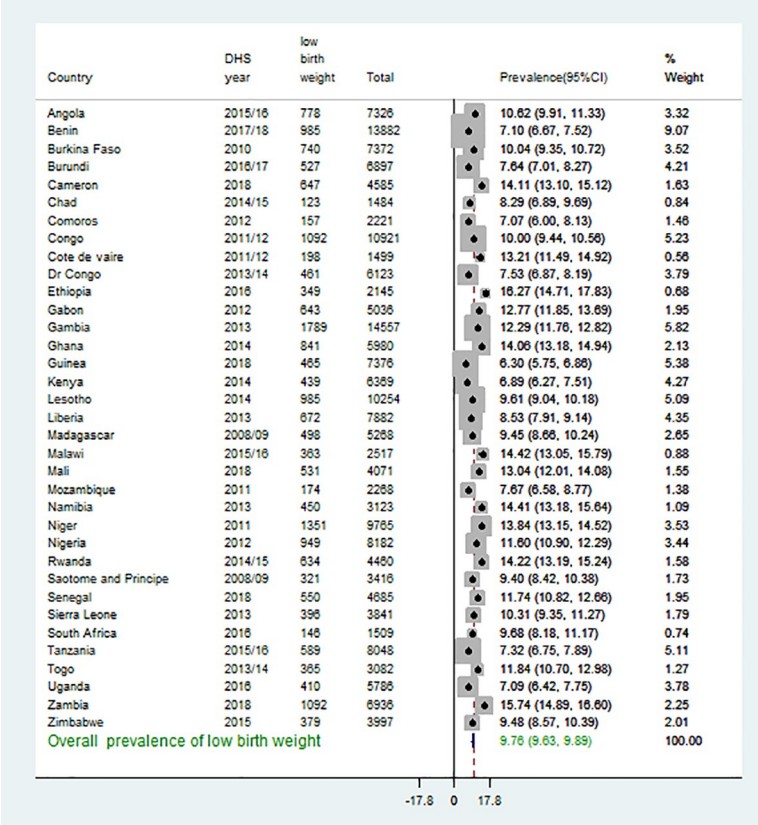

**Fig 2. Forest plot of pooled prevalence of low birth weight of new babies measured at birth in SSA.**

**Table 2. Model comparison and random effect results.**

| Parameter | Standard logistic regression | Mixed-effect logistic regression analysis (GLMM) |
|---|---|---|
| **LLR** | -45366 | -45363 |
| **Deviance** | 90732 | 90726 |
| **LR-test** | LR test vs. logistic model: chibar2(01) = 5.56 Prob > = chibar2 = 0.0092 | |
| **MOR** | 1.08 [1.05, 1.13] | |
| **Cluster variance** | 0.0069 [0.0025, 0.0169] | |

*LLR; log-likelihood ratio, ICC; Intra-class Correlation Coefficient, MOR; Median Odds Ratio, LR-test; Likelihood Ratio test.

**Table 3. Multivariable multilevel logistic regression analysis to assess factors associated with low birth weight in Sub-Saharan Africa.**

| Variables | Newborn birth weight (n = 202,878) | | Column (%) | Odds ratio | |
|---|---|---|---|---|---|
| | Normal and above | Low birth weight | | COR (95%CI) | AOR (95%CI) |
| **Residence** | | | | | |
| Urban | 75834 | 8798 | 84632(41.7) | 1 | |
| Rural | 106038 | 12208 | 118246(58.3) | 1.00(0.97,1.03) | 0.98(0.97,1.02) |
| **Maternal age** | | | | | |
| 15–19 | 10688 | 1753 | 12441(6.13) | 1 | 1 |
| 20–29 | 92530 | 10955 | 103485(51.01) | 0.71(0.68,0.75) | 0.75(0.71,0.80)* |
| 30–39 | 64427 | 6686 | 71113(35.05) | 0.62(0.59,0.65) | 0.76(0.71,0.82)* |
| 40–49 | 14226 | 1612 | 15838(7.81) | 0.66(0.62,0.71) | 0.86(0.78,0.95)* |
| **Mothers education status** | | | | | |
| No | 46045 | 6096 | 52141(25.70) | 1 | 1 |
| Primary | 68894 | 7811 | 76705(37.81) | 0.87(0.84,0.90) | 0.87(0.83,0.92)* |
| Secondary and above | 66935 | 7097 | 74052(36.49) | 0.81(0.78,0.90) | 0.79(0.75,0.84)* |
| **Husband education status** | | | | | |
| No | 36156 | 4795 | 40951(23.52) | 1 | 1 |
| Primary | 49617 | 5389 | 55006(31.60) | 0.82(0.79,0.86) | 0.89(0.84,0.95)* |
| Secondary and above | 70862 | 7276 | 78138(44.88) | 0.87(0.83,0.90) | 0.89(0.84,0.94)* |
| **Marital status** | | | | | |
| Married | 153994 | 17115 | 171109(84.34) | 1 | 1 |
| Divorced/widowed/not in union | 27877 | 3890 | 27787(15.66) | 1.27(1.23,1.32) | 1.22(1.16,1.28)* |
| **Wealth status** | | | | | |
| Poor | 61288 | 7600 | 68888(33.96) | 1 | 1 |
| Middle | 36022 | 4119 | 40141(19.79) | 0.91(0.87,0.95) | 0.91(0.86,0.95)* |
| Rich | 84563 | 9285 | 93848(46.26) | 0.83(0.81,0.86) | 0.82(0.78,0.86)* |
| **Women had occupation** | | | | | |
| No | 61442 | 7859 | 69303(34.16) | 1 | 1 |
| Yes | 120429 | 13145 | 133574(65.84) | 0.84(0.81,0.86) | 0.88(0.84,0.91)* |
| **Women participating in making their own health care decisions** | | | | | |
| Yes | 93000 | 9685 | 102685(50.61) | 1 | 1 |
| No | 88873 | 11320 | 100193(49.39) | 1.24(1.20,1.28) | 1.25(1.21,1.29)* |
| **Media exposure** | | | | | |
| No | 46390 | 5834 | 52224(25.74) | 1 | 1 |
| Yes | 135482 | 15172 | 105654(74.26) | 0.91(0.88,0.94) | 0.97(0.93,1.01) |
| **Number of ANC visit** | | | | | |
| No visit | 52323 | 7194 | 59518(29.34) | 1 | 1 |

*(Continued)*

**Table 3.** (Continued)

| Variables | Newborn birth weight (n = 202,878) | | Column (%) | Odds ratio | |
|---|---|---|---|---|---|
| | Normal and above | Low birth weight | | COR (95%CI) | AOR (95%CI) |
| 1–3 visits | 42443 | 5263 | 47706(23.51) | 0.90(0.87,0.94) | 0.91(0.83,0.99)* |
| ≥ 4 visits | 87105 | 8547 | 95653(47.15) | 0.71(0.69,0.74) | 0.72(0.66,0.79)* |
| **Preceding birth interval (in months)** | | | | | |
| < 24 | 22107 | 2320 | 24427(12.04) | 1 | 1 |
| ≥ 24 | 159765 | 18685 | 178450(87.96) | 1.09(1.04,1.14) | 1.09(1.04,1.14)* |
| **Parity** | | | | | |
| 1–2 | 83996 | 10803 | 94799(46.73) | | 1 |
| 3–5 | 69056 | 7098 | 76154(37.54) | 0.76(0.72,0.79) | 0.74(0.70,0.77)* |
| ≥6 | 28819 | 3104 | 31923(15.74) | 0.69(0.65,0.74) | 0.68(0.63,0.73)* |
| **Sex of child** | | | | | |
| Male | 93333 | 9494 | 102828(50.68) | 1 | 1 |
| Female | 88539 | 11510 | 100050(49.32) | 1.35(1.25,1.37) | 1.30(1.25,1.35)* |
| **Type of births** | | | | | |
| Single | 177599 | 17731 | 195330(96.28) | 1 | 1 |
| Twin | 4273 | 3274 | 7547(3.72) | 8.45(7.42,9.75) | 9.03(8.37,9.75)* |
| **Iron supplementation** | | | | | |
| No | 17715 | 1972 | 19687(13.28) | 1 | 1 |
| Yes | 116435 | 1248 | 128924(86.75) | 0.10(0.92,1.05) | 0.99(0.94,1.04) |

* P-value ≤0.05; COR = Crude Odds Ratio; AOR = Adjusted Odds Ratio; ANC = Antenatal care.

from clusters with lower birth weight. Besides, the likelihood ratio test was (LR test vs. Logistic model: chibar2 (01) = 5.56 Prob > = chibar2 = 0.0092), which informed that the mixed-effect logistic regression model (GLMM) is the better model over the basic model (Table 3).

## Determinants of low birth weight in Sub-Saharan Africa

In the multivariable mixed-effect logistic regression analysis; maternal age, maternal education, husband education, marital status, wealth status, maternal occupation, women decision making autonomy, number of ANC visit, birth interval, parity, sex of the child, and types of birth were significant predictors of LBW babies in Sub-Saharan Africa.

The odds of LBW among mothers age 20–29, 30–39 and 40–49 were decrease by 25% (AOR = 0.0.75, 95% CI: 0.71, 0.80), 24% (AOR = 0.076, 95% CI: 0.71, 0.82), and 14% (AOR = 0.86, 95% CI: 0.78, 0.95) than mothers aged 15–19 years, respectively. Mothers who attained primary education (AOR = 0.87, 95% CI: 0.83, 0.92) and secondary education and above (AOR = 0.79, 95% CI: 0.75, 0.84) were less likely to LBW at birth than mother who didn't attain formal education. Mothers whose husbands attained primary education (AOR = 0.89, 95% CI: 0.84, 0.94) and secondary education and above (AOR = 0.89, 95% CI: 0.84, 0.95) were less likely to LBW at birth than mother whose husband didn't attain formal education. Mothers who are Divorced/widowed/not living together were 1.22 times more likely LBW at birth than married mothers (AOR = 1.22, 95% CI: 1.16, 1.28). The odds of LBW among mothers of middle were decrease by 9% (AOR = 0.91, 95% CI: 0.86, 0.95) and 18% (AOR = 0.82, 95% CI: 0.78, 0.86) respectively. Mothers who had occupation decreased the odds of LBW at birth by 12% (AOR = 0.88, 95% CI: 0.84, 0.91) compared to mothers who had no occupation. Mothers who have no participation in making their own health care decisions

were 1.25 times more likely to LBW at birth than mothers who have no participation in making their own health care decisions (AOR = 1.25, 95% CI: 1.21, 1.29). Mothers who had 1–3 and four and above ANC visits were decreasing the odds of LBW at birth by 9% (AOR = 0.91, 95% CI: 0.83, 0.99) and 28% (AOR = 0.72, 95% CI: 0.66, 0.79) respectively. The odds of LBW of babies at birth among women who had 3–5 births and $\geq$ six births were decreased by 26% (AOR = 0.74, 95% CI: 0.70, 0.77) and 32% (AO = 0.68, 95% CI: 0.63, 0.73) than mothers who had 1–2 births respectively. Being a female baby increases the odds of LBW at birth by 30% as compared to males (AOR = 1.30, 95% CI: 1.25, 1.35). Being twin birth was 9.03 times more likely to LBW at birth than single birth (AOR = 9.03, 95% CI: 8.37, 9.75) (**Table 3**).

## Discussion

The pooled analysis from 35 SSA countries showed that the overall prevalence of LBW was 9.47%, with variations among African regions ranged from 8.9% in eastern Africa to 10.3% in Southern Africa. In addition, variations also observed between countries ranged from 6.3% in Guinea to 16.21% in Ethiopia, as shown in Fig 1. This was consistent with the results of systematic and meta-analysis findings in Iran (9%) [20], Nepal (9.4%) [19], and Ghana (9.69%) [21]. However, this work finding was lower than those studies in Kenya (12.3%) [12], selected African countries such as Burkina Faso, Ghana, Malawi, Senegal, and Uganda was, respectively, 13.4%, 10.2%, 12.1%, 15.7% and 10% [22], and systematic and meta-analysis results in Ethiopia (18%) [16]. This could be attributed to differences in study population and design. Demography and health surveys were conducted in the community-based study, whereas those studies included in the systematic and meta-analysis were single-center and health facility-based study that could overestimate the magnitude of LBW. In addition, the finding of this study was lower than global and sub-Saharan countries estimate of 14.6% and 16.4%, according to the national, regional, and global systematic analysis [4]. This could be because interventions and efforts made to curb maternal and neonatal mortality in low-income countries included in SSA [23]. Moreover, the overall prevalence of LBW in this study was higher than in a study elsewhere (7.6%) [24]. This could be due to socio-demographic, economic, health care system differences.

Different individual and contextual factors attributed to the occurrence of LBW, as the age of the mother increased the odds of low birth weight were decreased. This finding was consistent with those of studies conducted in Nepal and Iran [24–26]. This could be attributed to problems of child marriage and malnutrition in adolescent girls in Africa [27]. Similarly, some women and husbands' education levels are associated with decreased odds of LBW than uneducated women and men. This finding was consistent with those of studies in Ethiopia, Kenya, and Nepal [12,24–26,28]. Literacy is often associated with good knowledge about the better nutritional practice and health-seeking behaviors during pregnancies, which could impact birth outcomes [7].

Furthermore, middle, rich wealth status, those who had occupation associated with a reduced probability of LBW compared to the poor and those who had no occupation, respectively. This could be due to the fact women who had good economic status and occupation had better nutritional status during pregnancies [6,29]. In addition, better wealth conditions and occupation are more likely to increase women's health-seeking behaviors, which could be helpful for early detection of intrauterine growth retardations. This finding implies that improving women's socio-economic conditions might help reduce unfavorable birth outcomes like LBW. The odds of LBW for multiparous were reduced than primiparous women. This finding was consistent with those of studies [6,12]. Most of the pregnancy-related symptoms diminished as the parity and gravidity increased, which could impact the nutritional

status of women [19]. In addition, strengthening of focused ANC follow-up for primiparous mothers, counseling, and nutritional help.

In addition, one or more antenatal care (ANC) visits were associated with reduced odds of LBW compared to those who had no follow-ups. This could be due to the fact that ANC has packages like nutritional counseling and iron supplementations for maternal and fetal well-being. Conversely, divorced and separated women are associated with an increased risk of LBW compared to married women. This could be due to the fact that women divorced and had no partner support make them more vulnerable to malnutrition [7,11]. In addition, divorced and separated women have poor psychological and social support, which could impact adverse birth outcomes. Moreover, a mother who has poor marital relationships may experience intimate partner violence that leads to premature rupture of membrane and onset of preterm labor [30]. Addressing the social, psychological constraints, and financial hardship might have valuable importance to reducing unfavorable birth outcomes like LBW.

This study also showed that twin pregnancy is associated with higher odds of LBW than a singleton. This finding was consistent with those of studies in Lusaka, Zambia, and Nigeria [31,32]. This could be due to the fact that twin pregnancies are associated with the increased demand for nutrients and oxygenated blood [24]. Women who didn't participate in health-care-associated decision-making with increased LBW occurrence to their newborns compared to those who didn't participate in decision-making. Empowerment and women's autonomy would be helpful to make the right choices to access quality health care during pregnancy. Increasing women empowerment and autonomy to increase utilization of maternal health services. This finding was supported by previous results in Ghana [33] and Bangladesh [18]. Likewise, female newborns associated higher odds of LBW than males. This finding was consistent with those of a previous study [34]. Furthermore, wider birth intervals (above 24 months) are associated with an increased occurrence of LBW. This finding was contrary to a previous study in Ethiopia [34]. This could be due to the sample size differences and socio-demographic characteristics in SSA.

This study showed that iron supplementation did not create a statistically significant association with LBW during the prenatal era. However, the coefficient showed that the correlation between low birth weight and increased iron supplementation during the pregnancy period was negative. This is of clinical significance for both pregnant mothers and newborns and highlights the effects of improving the coverage of iron supplementation in pregnant women [35]. In addition to clinical evaluations, provision of therapy, and recognition of danger signs, iron supplementation is also an effective intervention during follow-up of antenatal treatment for pregnant mothers.

## Strength

This study was based on large observation (sample size) from SSA countries and helpful for an overall assessment of LBW. In addition, pooled estimates of LBW will be helpful for policy-makers and stakeholders for the situation of the problems. This result would be important for employing combined efforts to identified modifiable risk factors.

## Limitation

The data's cross-sectional nature may not show a true temporal relationship between the outcome and independent variables. Much of the birth weight was taken from the mother's self-report, and this may result in recall bias. The health section and calendar age/date boundaries, using data for last-born children or non-last-born children instead of all children born in the five years preceding the survey, will probably result in biased estimates [36]. Missing important

maternal clinical parameters (chronic illnesses status, malaria infection, nutritional status) were the limitations of the study. The problem is overfitting may affect our result. Climate changes such as heatwave exposure and air temperature rise are very prevalent in low-income countries; the impact on adverse birth outcomes such as low birth weight has not been assessed. Combined low birth weight projections can be useful for problem situations for policymakers and stakeholders.

## Conclusion

In this study, the magnitude of low birth weight was high with significant variations among countries. Female child, women not participated in healthcare decision making, and wider birth intervals, divorced/ separated women, and twin pregnancies associated with increased occurrences of low birth weight, whilst some level of woman and husband education, antenatal care visits, older maternal age, and multiparity associated with reduced occurrence low birth weight. This finding suggests that more emphasis is important for women with a lack of support, multiples, and addressing healthcare decision-making problems.

## Author Contributions

**Conceptualization:** Zemenu Tadesse Tessema, Koku Sisay Tamirat, Achamyeleh Birhanu Teshale, Getayeneh Antehunegn Tesema.

**Data curation:** Zemenu Tadesse Tessema, Koku Sisay Tamirat, Achamyeleh Birhanu Teshale, Getayeneh Antehunegn Tesema.

**Formal analysis:** Zemenu Tadesse Tessema, Koku Sisay Tamirat, Achamyeleh Birhanu Teshale, Getayeneh Antehunegn Tesema.

**Funding acquisition:** Zemenu Tadesse Tessema, Koku Sisay Tamirat, Achamyeleh Birhanu Teshale, Getayeneh Antehunegn Tesema.

**Investigation:** Zemenu Tadesse Tessema, Koku Sisay Tamirat, Achamyeleh Birhanu Teshale, Getayeneh Antehunegn Tesema.

**Methodology:** Zemenu Tadesse Tessema, Koku Sisay Tamirat, Achamyeleh Birhanu Teshale, Getayeneh Antehunegn Tesema.

**Project administration:** Zemenu Tadesse Tessema, Koku Sisay Tamirat, Achamyeleh Birhanu Teshale, Getayeneh Antehunegn Tesema.

**Resources:** Zemenu Tadesse Tessema, Koku Sisay Tamirat, Achamyeleh Birhanu Teshale, Getayeneh Antehunegn Tesema.

**Software:** Zemenu Tadesse Tessema, Koku Sisay Tamirat, Achamyeleh Birhanu Teshale, Getayeneh Antehunegn Tesema.

**Supervision:** Zemenu Tadesse Tessema, Koku Sisay Tamirat, Achamyeleh Birhanu Teshale, Getayeneh Antehunegn Tesema.

**Validation:** Zemenu Tadesse Tessema, Koku Sisay Tamirat, Achamyeleh Birhanu Teshale, Getayeneh Antehunegn Tesema.

**Visualization:** Koku Sisay Tamirat, Achamyeleh Birhanu Teshale, Getayeneh Antehunegn Tesema.

**Writing – original draft:** Zemenu Tadesse Tessema, Koku Sisay Tamirat, Achamyeleh Birhanu Teshale, Getayeneh Antehunegn Tesema.

**Writing – review & editing:** Zemenu Tadesse Tessema, Koku Sisay Tamirat, Getayeneh Antehunegn Tesema.

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
