## [Decision Letter · Decision Letter 0]

2 Dec 2020

PONE-D-20-27726

Determinants of low birth weight at birth in Sub-Saharan Africa: A Generalized Linear Mixed   Model

PLOS ONE

Dear Dr. Tadesse,

Thank you for submitting your manuscript to PLOS ONE. After careful consideration, we feel that it has merit but does not fully meet PLOS ONE’s publication criteria as it currently stands. Therefore, we invite you to submit a revised version of the manuscript that addresses the points raised during the review process.

All the reviewers recognise some merit in this manuscript but also have some concerns about it. Each of them have highlighted changes that need to be made in any revision.

I would add the following points:

1. The standard of English used in the manuscript needs to be improved (maybe via checking by a native English speaker).

2. Please check that the manuscript is consistent with the RECORD reporting checklist (http://record-statement.org/checklist.php). At the moment there are no details of how to access the study protocol or programming code as one example where the manuscript is not currently fully compliant.

I look forward to seeing a revised version of the manuscript.

We look forward to receiving your revised manuscript.

Kind regards,

Clive J Petry, PhD

Academic Editor

PLOS ONE

Journal Requirements:

5. We noticed you have some minor occurrence of overlapping text with the following previous publications, which needs to be addressed:

- https://worldwidescience.org/topicpages/l/lbw+births+lbw.html

- http://etd.aau.edu.et/bitstream/handle/123456789/8229/29.Andenet%20Gebrekidan.pdf?sequence=1

In your revision ensure you cite all your sources (including your own works), and quote or rephrase any duplicated text outside the methods section. Further consideration is dependent on these concerns being addressed.

Reviewers' comments:

Reviewer's Responses to Questions

**Comments to the Author**

1. Is the manuscript technically sound, and do the data support the conclusions?

Reviewer #1: Partly

Reviewer #2: No

Reviewer #3: Yes

2. Has the statistical analysis been performed appropriately and rigorously? 

Reviewer #1: Yes

Reviewer #2: No

Reviewer #3: No

3. Have the authors made all data underlying the findings in their manuscript fully available?

Reviewer #1: Yes

Reviewer #2: Yes

Reviewer #3: Yes

4. Is the manuscript presented in an intelligible fashion and written in standard English?

Reviewer #1: Yes

Reviewer #2: Yes

Reviewer #3: Yes

5. Review Comments to the Author

Reviewer #1: The authors were wise to take into account the between cluster variability by using the appropriate models. Therefore, a mixed effect logistic regression model (both fixed and random effect) was fitted. Since the outcome variable was binary, standard logistic regression and Generalized Linear Mixed Models (GLMM) were fitted, as expected.

Assuming the model inputs are appropriate this is a straight forward application of the models. The timing of the information is seen in Table 1 as noted in the ‘Data Source’ information. Also assurances should be given of the homogeneity of the sample across the countries.

Reviewer #2: The authors present an important analysis assessing risk factors for low birthweight in Sub-Saharan Africa. The analysis pools recent (2008-2018) DHS surveys from 35 different to represent the sub-continent and specific regions, Central Africa, East Africa, West, Africa, South Africa. The authors find risk factors and protective factors for low birth weight similar to those established in the literature. While there may be some advantage to assessing risk factors by region as the authors aim, there are some concerns with the overall methods used in the manuscript. Further the manuscript would benefit from English language editing. I have provided some suggestions below. See the attachment for more details.

Reviewer #3: The topic of determinants of low birth weight at birth is of high interest, particularly in sub-Saharan Africa. The manuscript is clearly written and presented.

However, there are some aspects that the authors should consider.

Major comments

1. The principal issue of this work is the use of DHS data.

DHS is known as a formidable source of data and give a general overview of sanitary situation of each country (trend of key indicators). However, there are many methodological concerns about accuracy of data collection, population sample and high risk of potential biases: Selection biases (due to different household sampling techniques), information or classification biases (due to self-report data or declared data, medical records, standardization of data collection …). In addition, the methodology can differ between countries, especially concerning the under-five children data. Indeed, two main different methods can be used to collect these data: Most-Recently-Born Children and Non-Most-Recently-Born Children.

I invite the authors to read this critical analysis of the use of DHS data for children (pdf attached), whose conclusion I quote: “When researchers use the DHS surveys for the analysis of children born in the five years preceding the survey, they must be very careful to consider and avoid inherent biases that derive from both the data structure and the nature of the analysis. Beyond the usual considerations of omission of births and transference of births across questionnaire age boundaries, particularly the health section and calendar age/date boundaries, using data for last-born children or non-last-born children instead of all children born in the five years preceding the survey will probably result in biased research findings.”

2. Analysis (statistical issue)

Multiple pregnancies (such as twin pregnancy) were included in the same models with single pregnancy. Multiple pregnancies (MP) are known as a mediator factor of LBW (MP is on the pathway of LBW). Therefore, this can lead to a sur-adjustment. I will be more useful to perform the main analysis after excluding MP and to perform a sensitivity analysis with MP.

Other comments

The use of “Determinants” in the title by the authors is misleading to me. The notion of determinant implies a causal role while the notion of risk factor or associated factor is broader and refers to a higher probability of disease in exposed subjects. In this work, it is rather “associated factors” that have been highlighted.

Abstract

Lines 47-48: In conclusion of abstract, the authors declared “this study… with significant variations among countries” but any variations or range have been presented in the results paragraph of abstract. Please, correct it.

Introduction

Lines 71-72: 25-30 times seems very high today (substantial progress has been made, even in countries with limited resources). I suggest that the authors further clarify this information, either by adding that this very high risk is for very low birth weight infants, or that the information is several years old.

Lines 80-82: There are many studies such as systematic reviews, multicenter pooled analyses (some of which have even been cited by the authors). The rationale for this study therefore needs to be better specified. For the moment, we do not really see the added value or the new knowledge brought.

Lines 85-86: I am not convinced.

Methods

See my major comment about the use of DHS data. I suggest that the authors briefly present the methods used to collect data on children in the DHS to give us an idea of the robustness or otherwise of the data and potential biases.

Also, what about children who were not included in this analysis. Do the authors check any potential selection biases?

It will be useful to have a country included in each region.

How infant weight were measured at birth? Standardization? Declared (or self-report data?) Medical record?

Variable definition paragraph is missing in this work, particularly for the clarification of the definition of variable such as “household wealth index”, “women health care decision making autonomy”.

What about other important independent variables such as malaria infection or BMI?

ANC: please, define the acronym the first time you use it.

The parameters of mixed logistic regression need to be more detail. Indeed, the authors used multilevel models to take into account the hierarchical structure of data. However, it is important to specify what represents the 1st level (individual?), the 2nd level (regions or countries?). Please, clarify.

Results

Table 2: West Africa weighted frequency is 12170, but Benin (which is West African country) only provide 13909 participants. Please, clarify.

Table 4: It will be useful to have also percentage in normal and LBW columns.

Lines 162-163: “Women from a cluster … lower birth weight”. This sentence is a bit confusing. Please, clarify.

Discussion

Line 197: “8.9% to 10.3%”, please include this result in results session also.

In general, the discussion section need to be more focus on comparison of this study with other multicentric pooled analysis and meta-analysis studies in sub-Saharan Africa, instead of comparison of single center or no-pooled analysis studies.

Lines 199-207: For me, these comparisons are not relevant (single center studies vs. pooled multicenter study).

Lines 214-224: All this paragraph is a speculation. What about nutritional factors in this study. Iron supplementation was not associated with LBW in this study. Why? A discussion of this negative finding will be useful. Why didn't the authors consider other indicators of nutritional status such as BMI? If nutritional status plays an important role in the occurrence of LBW, all of the above should be discussed.

See in major comment my comment about multi pregnancies.

Limitations paragraph need to be more detail, including discussion about potential methodological issues, need to take into account nutritional data and the robustness of the findings.

6. PLOS authors have the option to publish the peer review history of their article (what does this mean?). If published, this will include your full peer review and any attached files.

Reviewer #1: No

Reviewer #2: **Yes: **Emily Deichsel

Reviewer #3: No

---

## [Author Response · Author response to Decision Letter 0]

29 Dec 2020

PLOS ONE 

Point by point response for editors/reviewers comments 

The manuscript title “Prevalence of low birth weight and its associated factor at birth in Sub-Saharan Africa: A Generalized Linear Mixed Model”

Manuscript ID: PONE-D-20-27726 

Dear editor/reviewer. 

Dear all,

I would like to thank you for these fair, constructive, building, and improvable comments on this manuscript that would improve the substance and content of the manuscript. We considered each comment and clarification questions of reviewers on the manuscript thoroughly. My point-by-point responses for each comment and question are described in detail on the following pages. Further, the details of changes were shown by track changes in the supplementary document attached.

Response to Reviewers comments 

Reviewer #1

1. The timing of the information is seen in Table 1 as noted in the ‘Data Source’ information. Also assurances should be given of the homogeneity of the sample across the countries

Author response:- Thank you very much for your fair and contractive comments. We accept your comment and corrected it accordingly. We included your concern in the revised manuscript which describe the homogeneity of sampling procedure across sub-Saharan Africa countries line 94-95(see the revised manuscript)

Reviewer #2

1. There may be some advantage to assessing risk factors by region as the authors aim, 

Author’s response:- Thank you very much for your concern. There are different research articles done on assessing risk factors by regions(Eastern Africa, Southern Africa, Western Africa, and Central Africa). Our primary objective was assessing factors that represent and can be generalizable at sub-Saharan Africa which consists of the entire four WHO sub-Saharan Africa region(Eastern Africa, Southern Africa, Western Africa, and Central Africa). The 35 sub-Saharan Africa country datasets were appended together for the prediction of low birth weight babies in SSA. The DHS data is a nationally representative survey that collects data on basic health indicators like mortality, morbidity, family planning service utilization, fertility, maternal and child health. The data were derived from the measure DHS program. The DHS has different datasets (men, women, children, birth, and household datasets). For this study, we used the children dataset(KR dataset.)

2. There are some concerns with the overall methods used in the manuscript. 

Authors response:- Thank you very much for your concern. We would be very happy if the comment was clear and indicate the specific methodological problem. The given comment is crude and blind. We would be very happy if you tell us the methodological problems you observed. The method session is adopted from the procedure of DHS program from across all sub-Saharan Africa countries. We try to use the advanced statistical technique that considers the hierarchical nature of the DHS data i.e Generalized Linear Mixed Effect Model(GLMM). We are open to correct the methodological problems you observed if you indicate us clearly for the improvement of the manuscript. 

3. The manuscript would benefit from English language editing. I have provided some suggestions below. See the attachment for more details

Authors response:- Thank you very much for your contractive and fair comments for the improvement of the manuscript. We accept your comment and corrected it accordingly. In addition, we gave our manuscript to an English editor and we corrected grammatical and punctuation errors accordingly (see the revised manuscript).

Reviewer #3

1. The principal issue of this work is the use of DHS data.

DHS is known as a formidable source of data and give a general overview of sanitary situation of each country (trend of key indicators). However, there are many methodological concerns about accuracy of data collection, population sample and high risk of potential biases: Selection biases (due to different household sampling techniques), information or classification biases (due to self-report data or declared data, medical records, standardization of data collection …). In addition, the methodology can differ between countries, especially concerning the under-five children data. Indeed, two main different methods can be used to collect these data: Most-Recently-Born Children and Non-Most-Recently-Born Children.

I invite the authors to read this critical analysis of the use of DHS data for children (pdf attached), whose conclusion I quote: “When researchers use the DHS surveys for the analysis of children born in the five years preceding the survey, they must be very careful to consider and avoid inherent biases that derive from both the data structure and the nature of the analysis. Beyond the usual considerations of omission of births and transference of births across questionnaire age boundaries, particularly the health section and calendar age/date boundaries, using data for last-born children or non-last-born children instead of all children born in the five years preceding the survey will probably result in biased research findings.

 Authors response:- Thank you very much for the concerned reviewer. For this study, we used KR dataset. This dataset records children with the most recent one rather than all children in the household. We read a pdf you attached us. We included this issue as a limitation after we read the pdf. For this study, we used the most recent birth within five years preceding the survey for each country. Therefore, not to miss leading the reader we included it in the limitation session(see the revised manuscript).

Analysis (statistical issue)

2. Multiple pregnancies (such as twin pregnancies) were included in the same models as a single pregnancy. Multiple pregnancies (MP) are known as a mediator factor of LBW (MP is on the pathway of LBW). Therefore, this can lead to a sur-adjustment. I will be more useful to perform the main analysis after excluding MP and to perform a sensitivity analysis with MP.

Authors response:- We conducted the sensitivity analysis according to your recommendation(with and without the multiple pregnancies the result presented as below). 

The sensitivity analysis without MP(multiple pregnancies)

Sensitivity analysis with MP (multiple pregnancies ) included 

Model comparison With MP(multiple pregnancies) Without multiple pregnancies Decision

LL -45356 -46696 Largest bette

Deviance 90717 93392 Smallest better

Thank you very much for your concern. We conducted a sensitivity analysis with and without including multiple pregnancies in the model. The Area under the curve (ROC) curves with multiple pregnancies in the model was 62.19% and without Multiple pregnancies in the model was 58.18%. Therefore, including MP in the model is better than excluding it. The likelihood (Largest better)and Deviance(smallest better) also support model with MP included was better than the model without MP. The works of literature also support that multiple pregnancies are a predictor of low birth weight. 

3. The use of “Determinants” in the title by the authors is misleading to me. The notion of determinant implies a causal role while the notion of the risk factor or associated factor is broader and refers to a higher probability of disease in exposed subjects. In this work, it is rather “associated factors” that have been highlighted.

Authors response:- Thank you very much for your comment. We accept your comment and corrected it accordingly. We change the title to “Prevalence of low birth weight and its associated factor at birth in Sub-Saharan Africa: A Generalized Linear Mixed Model”(see the revised manuscript). 

Abstract

4. Lines 47-48: In conclusion of abstract, the authors declared “this study… with significant variations among countries” but any variations or range have been presented in the results paragraph of abstract. Please, correct it.

Authors response:- Thank you very much for your contractive comment. We accept your comment and corrected it accordingly line number 48-49 (see the revised manuscript).

Introduction

5. Lines 71-72: 25-30 times seems very high today (substantial progress has been made, even in countries with limited resources). I suggest that the authors further clarify this information, either by adding that this very high risk is for very low birth weight infants, or that the information is several years old.

Author’s response:- Thank you very much for your contractive comment. We accept your comment and corrected it accordingly line number 71 (see the revised manuscript).

6. Lines 80-82: There are many studies such as systematic reviews, multicenter pooled analyses (some of which have even been cited by the authors). The rationale for this study therefore needs to be better specified. For the moment, we do not really see the added value or the new knowledge brought.

Author’s response:- Thank you very much for your comment. We accept your comment and corrected it accordingly line number 80-87 (see the revised manuscript)

7. Lines 85-86: I am not convinced.

Authors response:- Thank you very much for your comment. The intention for this point was the finding of this study is better than systematic review and meta-analysis. The reason was DHS program used the same study design(cross-sectional nature), the same manual for data collection, the variable coding, the same sampling procedure etc… As we know in the systematic review and meta-analysis there may be different study design, data collection tool, different coding of variables, publication bias, etc…..because of this difference we believe that this study finding is better than a systematic review and meta-analysis. We remove this idea from the revised manuscript if it misses leads readers and we substitute other word line number 90-92(see the revised manuscript).

Methods

8. See my major comment about the use of DHS data. I suggest that the authors briefly present the methods used to collect data on children in the DHS to give us an idea of the robustness or otherwise of the data and potential biases.

Authors response:- Thank you very much for your concern. For this study, we used KR(kids record dataset). This dataset records the last or the recent child in the family. We read your materials attached us with the comment regarding data analysis, especially for under-five children data. We agree with your concern. There may be a bias result if we did not include all children’s in the family. According to our scenario, we used the most recent child five years preceding the survey for each 35 sub-Saharan Africa countries. So we boldly mentioned this point in the limitation session line (see the revised manuscript)

9. What about children who were not included in this analysis. Do the authors check any potential selection biases?

Authors response:- Thank you very much for your constructive comment. The data used for this study was secondary data obtained from DHS program after project proposal was submitted to the DHS program office and permission was obtained from data acrchivest. The dataset records for those children born from women within five years preceding the survey and the most recent one. Within one family there may be more than one child but the KR dataset recorded the most recent one. According to the above scenario, there is no selection bias for this study because the most recent bias was obtained. 

10. It will be useful to have a country included in each region.

Authors response:- Thank you very much for your contractive comment. We accept your comment and corrected it accordingly see Table 1 (see the revised manuscript ). 

11. How infant weight were measured at birth? Standardization? Declared (or self-report data?) Medical record?

Authors response:- Thank you very much for your comment. The birth size was measured using self-report from mother recall, medical records from a nearby health facility, and weights measured at birth. In SSA a small number of children were recorded their weight birth size. Most infant weights were obtained from their mother's self-report. Because of this, we include this point as a limitation because this may result recall bias.

12. Variable definition paragraph is missing in this work, particularly for the clarification of the definition of variables such as “household wealth index”, “women health care decision making autonomy”.

Authors response:- Thank you very much for your contractive comment. We accept your comment and corrected it accordingly line 130-137(see the revised manuscript ). 

13. What about other important independent variables such as malaria infection or BMI?

Authors response:- Thank you very much for your concern. Yes, you are right these variables are important independent variables but malaria infection was not collected in the DHS dataset and BMI has had a high missing value because of this we excluded it from the analysis.

14. ANC: please, define the acronym the first time you use it.

Authors response:- Authors response:- Thank you very much for your contractive comment. We accept your comment and corrected it accordingly(see the revised manuscript ). 

15. The parameters of mixed logistic regression need to be more detail. Indeed, the authors used multilevel models to take into account the hierarchical structure of data. However, it is important to specify what represents the 1st level (individual?), the 2nd level (regions or countries?). Please, clarify.

Authors response:- Thank you very much for your comment. There are two types of variable in the analysis. Level one varianles include (maternal education, husband education, maternal age, mother marital status, household wealth index, maternal occupation status, women health care decision making autonomy, media exposure, number of ANC visit, preceding birth interval, parity, sex of the child, type of birth and iron supplementation) and level two variables (residence , region and country)

Results

16. Table 2: West Africa weighted frequency is 12170, but Benin (which is West African country) only provide 13909 participants. Please, clarify.

Authors response:- Thank you very much for your comment. We corrected it accordingly(see the revised manuscript). 

17. Table 4: It will be useful to have also percentage in normal and LBW columns.

Authors response:- Thank you very much for your comment. We corrected it accordingly(see the revised manuscript)

18. Lines 162-163: “Women from a cluster … lower birth weight”. This sentence is a bit confusing. Please, clarify.

Authors response:- Thank you very much for your comment. We accept your comment and corrected it accordingly(see the revised manuscript)

Discussion

19. Line 197: “8.9% to 10.3%”, please include this result in results session also.

Authors response:- Thank you very much for your comment. We corrected it accordingly(see the revised manuscript)

20. In general, the discussion section need to be more focus on comparison of this study with other multicentric pooled analysis and meta-analysis studies in sub-Saharan Africa, instead of comparison of single center or no-pooled analysis studies.

Authors response:-Thank you very much for your comment. We accept your comment and corrected it accordingly(see the revised manuscript). 

21. Lines 199-207: For me, these comparisons are not relevant (single center studies vs. pooled multicenter study).

Authors response:- Thank you very much for your comment. We accept your comment and corrected it accordingly(see the revised manuscript).

22. Lines 214-224: All this paragraph is a speculation. What about nutritional factors in this study. Iron supplementation was not associated with LBW in this study. Why? A discussion of this negative finding will be useful. Why didn't the authors consider other indicators of nutritional status such as BMI? If nutritional status plays an important role in the occurrence of LBW, all of the above should be discussed.

Authors response:- Thank you very much for your comment. We accept your comment and corrected it accordingly(see the revised manuscript).

23. See in major comment my comment about multi pregnancies.

Authors response:- Thank you very much for your comment. We try to address this issue in number two of your question.

24. Limitations paragraph need to be more detailed, including discussion about potential methodological issues, need to take into account nutritional data and the robustness of the findings.

Author response:- Thank you very much for your comment. We revised the limitation section of this manuscript after your suggestion including missed variables in the dataset and methodological issues that readers should take into account(see the revised manuscript).

---

## [Decision Letter · Decision Letter 1]

19 Jan 2021

PONE-D-20-27726R1

Prevalence of low birth weight and its associated factor at birth in Sub-Saharan Africa: A Generalized Linear Mixed   Model

PLOS ONE

Dear Dr. Tadesse,

Thank you for submitting your manuscript to PLOS ONE. After careful consideration, we feel that it has merit but does not fully meet PLOS ONE’s publication criteria as it currently stands. Therefore, we invite you to submit a revised version of the manuscript that addresses the points raised during the review process.

Although the revised manuscript is an improvement over the original version, reviewer 2 is concerned that you did not seem to get their comments - certainly the manuscript was not revised accordingly. Therefore we are giving you another chance to deal with them. Each point needs to be dealt with carefully in another revision (especially those marked as "major"). I look forward to seeing the next version.

We look forward to receiving your revised manuscript.

Kind regards,

Clive J Petry, PhD

Academic Editor

PLOS ONE

Reviewers' comments:

Reviewer's Responses to Questions

**Comments to the Author**

1. If the authors have adequately addressed your comments raised in a previous round of review and you feel that this manuscript is now acceptable for publication, you may indicate that here to bypass the “Comments to the Author” section, enter your conflict of interest statement in the “Confidential to Editor” section, and submit your "Accept" recommendation.

Reviewer #1: All comments have been addressed

Reviewer #2: (No Response)

2. Is the manuscript technically sound, and do the data support the conclusions?

Reviewer #1: (No Response)

Reviewer #2: No

3. Has the statistical analysis been performed appropriately and rigorously? 

Reviewer #1: (No Response)

Reviewer #2: No

4. Have the authors made all data underlying the findings in their manuscript fully available?

Reviewer #1: (No Response)

Reviewer #2: Yes

5. Is the manuscript presented in an intelligible fashion and written in standard English?

Reviewer #1: (No Response)

Reviewer #2: Yes

6. Review Comments to the Author

Reviewer #1: (No Response)

Reviewer #2: PLOS ONE

The authors present an important analysis assessing risk factors for low birthweight in Sub-Saharan Africa. The analysis pools recent (2008-2018) DHS surveys from 35 different to represent the sub-continent and specific regions, Central Africa, East Africa, West, Africa, South Africa. The authors find risk factors and protective factors for low birth weight similar to those established in the literature. While there may be some advantage to assessing risk factors by region as the authors aim, there are some concerns with the overall methods used in the manuscript. Further the manuscript would benefit from further English language editing. It appears the authors did not receive the detailed comments and suggestions below after the first review.

Major:

Methods

1) A pooled risk factor analysis may be inappropriate with these data. Pooling multiple surveys from 35 countries over a 10-year period results in a poorly defined population. An analysis such as LBW is likely to be overpowered with such a large dataset and so may detect statistical differences that are not clinically meaningful. For example, all of the included risk factors had a pvalue<0.2 and were included in the multivariable analysis. For low birthweight, you likely to have enough power in each individual survey and so don’t benefit from this type of pooling. It would be appropriate however to use weighted data from these studies to estimate relative burden of LBW and the risk factors in the larger region or sub-regions. The authors could consider focusing on just one region and stratifying by country and assess for differences by country.

2) It would be helpful for the authors to provide a flow chart of the data. It would be helpful to know what proportion of respondents were excluded based on missing birthweight and know if the risk factors were similar to those who were included in the study. Please include number of live births, and what proportion had LBW measured.

3) Line 101-102: Please provide DHS or other citations (or an appendix) referencing the sampling procedure in each country is the same. It is very likely that the sampling strategies were different in different countries.

4) Line 125: Please provide more details about the meta analysis shown in the forest plot. I assume this was fixed effects meta analysis? Is this how the pooled prevalence was calculated? Please clarify. The forest plot is only a visualization of the analysis.

5) It seems more levels of the hierarchical model may be needed. Consider clustering on the woman, household, cluster and country. (region might not be necessary).

6) It is not clear how missing data is handled. Was the multivariable model a complete case analysis?

Results:

7) Line: 166: the text says 202878 mothers were included, yet the variables in table 2 sub to 203,466. Is this due to the weighting? Other numbers in this paragraph also don’t match the tables.

8) Line 166: I think you mean West Africa and not East Africa?

9) Table 2 shows the same number of women and the same number of children. If all children born to women in the last 5 years are included in the analysis, I would expect there to be more children included in the analysis than women included. Please clarify the discrepancy and add the number of women and number of children included in the analysis separately in the text (the flow chart in suggestion 2 would also help this point).

10) Sample sizes from table 1 don’t match those listed in figure1

11) The variable of husband’s education is misrepresented. 31904 women who are not married should have husband’s education as missing.

12) Please clarify the meaning of “No” for education status. Do you mean less than primary education?

13) Similar to 6: Please check variable for preceding birth interval. How is this defined for a woman’s first birth? If they are in the ≥24 months, this could be the reason for the unexpected results.

14) Table 4: Please present column % instead of row % for comparison between normal and low birthweight.

Discussion:

15) Overall the discussion could be improved to put the results in context for the reader. Consider restricting the discussion to focus on some key results. The conclusion discusses modifiable risk factors. One approach could be to discuss the modifiable risk factors and the context of interventions for these risk factors. Brining in the discussion of preterm and fetal growth restriction may also bring context to these results.

16) All statements of explanation for these findings should be based on the literature and so need to be sited (line 227, 233, 237, 240, 249, 252, 254. 258).

17) Line 236: LBW associated with multiples is well established and it is an oversight here not to mention that multiples are almost always born early, and so increased risk for LBW for that reason.

Minor:

In response to reviewer 3’s suggestions, remove the word determinants in in line 38, 184

Introduction:

1) Line 56: Please define LBW abbreviation – the first sentence could read, Low birth weight (LBW) is one of the….

2) Line 61-63: The last sentence in the first paragraph doesn’t seem relevant to this overall summary that is at the SSA level. I suggest deleting or replacing high and low estimates for the region.

3) Line 71 “well facts” should be reworded. Could say are well established.

4) Line 78 Nutritional shouldn’t be capitalized.

5) Please provide citations for other LBW studies, meta analyses systematic reviews that referenced in line 83.

6) 89-92. I am not convinced region level estimates are very informative for interventions and policy decisions.

Methods:

7) Line 96-101: Please include rational for why these countries were selected. The list does not include the WHO African region or UN Africa region.

8) Line 111: the specific coded values is not necessary to include. This line could just say “LWB was defined as a birth weight <2.5kg, and those ≥2.5kg were considered normal and above normal birthweights.” – this suggestion could be applied throughout the manuscript.

9) Line 122: “Based on kinds of literature” needs to be edited. It could be revised to say, potential risk factors for LBW were included based on a review of the literature. (Please include a citation).

10) Line 156: If the P-value is used to base the decision on inclusion into the multivariable model, please include P-values with the bi-variate results. Because of the study is over powered for these detections, consider lowering the threshold of what is included in the multivariable analysis. If there were variables considered that did not meet this threshold, they should also be included in the manuscript.

Results:

1) Line 172: Please reword the description of highest low birth weight and lowest birthweight to make it easier to read. Consider saying highest LBW prevalence, or lowest rate of LBW births.

2) Line 193: Careful saying mothers were less likely to low birthweight. Consider revising to say “were less likely to deliver LBW infants than mothers without formal education:

3) Line 201: type on the confidence interval. Should be 0.91 instead of 0.81 (according to the table)

4) Line 210: Typo Bing should be being.

5) Table 2: please define media exposure.

6) Table 4 CORs are missing on the last page for parity, sex of child, type of births, iron supplementation.

Discussion

7) Line 229: The statement about Brazil seems irrelevant for a paper focused on SSA- consider removing. The authors could consider comparing to other regions instead.

8) consider the term multiples throughout instead of multiple pregnancies.

Considerations

1) It would be helpful to re-orient the risk factors so they are all risk factors or protective factors to help readability.

2) Consider looking for trends in birthweight by year, may want to control for the year of the DHS survey.

3) Consider combining Table 2 and 4.

7. PLOS authors have the option to publish the peer review history of their article (what does this mean?). If published, this will include your full peer review and any attached files.

Reviewer #1: No

Reviewer #2: **Yes: **Emily L Deichsel

---

## [Author Response · Author response to Decision Letter 1]

15 Feb 2021

PLOS ONE 

Point by point response for editors/reviewers comments 

The manuscript title “Prevalence of low birth weight and its associated factor at birth in Sub-Saharan Africa: A Generalized Linear Mixed Model”

Manuscript ID: PONE-D-20-27726R1 

Dear editor/reviewer. 

Dear all,

I would like to thank you for these fair, constructive, building, and improvable comments on this manuscript that would improve the substance and content of the manuscript. We considered each comment and clarification questions of reviewers on the manuscript thoroughly. My point-by-point responses for each comment and question are described in detail on the following pages. Further, the details of changes were shown by track changes in the supplementary document attached.

Response to Reviewers comments 

Reviewer #1

Major: 

Methods 

1) A pooled risk factor analysis may be inappropriate with these data. Pooling multiple surveys from 35 countries over a 10-year period results in a poorly defined population. An analysis such as LBW is likely to be overpowered with such a large dataset and so may detect statistical differences that are not clinically meaningful. For example, all of the included risk factors had a pvalue<0.2 and were included in the multivariable analysis. For low birthweight, you likely to have enough power in each individual survey and so don’t benefit from this type of pooling. It would be appropriate however to use weighted data from these studies to estimate relative burden of LBW and the risk factors in the larger region or sub-regions. The authors could consider focusing on just one region and stratifying by country and assess for differences by country. 

Author’s Response:- Thank you very much for your comment. Our objective was to know the pooed magnitude of LBW and factors related to Low Birth Weight (LBW) at the SSA level, not a single region (East, West, Central, and South). Thank you very for your concern related to overfitting. Actually having an adequate sample size gives an efficient estimate and robust standard error. But the problem is overfitting. After your comment, we included this issue in our limitation session(line 300) as it may affect our result. If you know any solution for such types of problems (overfitting) we are ready to learn from you. For this study, we used weighted data for analysis(table 2 weighted frequency done using weighting variable v005). 

2) It would be helpful for the authors to provide a flow chart of the data. It would be helpful to know what proportion of respondents were excluded based on missing birthweight and know if the risk factors were similar to those who were included in the study. Please include the number of live births, and what proportion had LBW measured. 

Author’s response:-Thank you very much for your nice and constructive comments. We accept your comment and included a flow chart of the data. A total of 365,818 live births were recorded in the 35 SSA countries. From this only 202, 878(55%) births were their birth weight measure at birth. 162,940 of them were excluded since their weight was not measured at birth and missingness. 

3) Line 101-102: Please provide DHS or other citations (or an appendix) referencing the sampling procedure in each country is the same. It is very likely that the sampling strategies were different in different countries. 

Author’s response:-Thank you very much for your comment. We accept your comment and we included the citation in the main text. The DHS program uses the same variable code, data collection manual, and sampling procedure though out the world that’s why we appended each country by giving code for each and able to work this analysis. You can read more on (https://www.dhsprogram.com/pubs/pdf/DHSG1/Guide_to_DHS_Statistics_DHS-7_v2.pdf ). 

4) Line 125: Please provide more details about the meta analysis shown in the forest plot. I assume this was fixed effects meta analysis? Is this how the pooled prevalence was calculated? Please clarify. The forest plot is only a visualization of the analysis. 

Author’s response:-Thank you very much for your constructive comments. We accept your comments and included the meta-analysis part in the method session of data management and analysis section line 156-158 (see the revised manuscript)

5) It seems more levels of the hierarchical model may be needed. Consider clustering on the woman, household, cluster and country (region might not be necessary). 

Author’s Response:- Thank you very much for your comment. Regarding the levels of a variable actually, some literature uses more than two levels. For our study, we included two levels(individual and community indicated in the independent variable ). We included region (SSA) as a community level variable because we believe that there is a variation on LBW in each sub-regions(South, East, West, and Central Africa). The forest plot result supports this. 

6) It is not clear how missing data is handled. Was the multivariable model a complete case analysis? 

Author’s Response:- Thank you very much for your comment. The missing data concerning the outcome(Birth weight(normal vs LBW)) variable was the drop. Not only missing data but also those births with which birth weight not measured at birth were also drop before any statistical analysis (see the revised manuscript in the flow chart of the data). 

Results:

7) Line: 166: the text says 202878 mothers were included, yet the variables in table 2 sub to 203,466. Is this due to the weighting? Other numbers in this paragraph also don’t match the tables. 

Author’s Response:- Thank you very much for your constructive comments. Due to tabulating errors in Stata, some variables were presented without weighing it. We accept your comment and corrected the table figures accordingly after weighing the figure 202,878 is the weighted total sample size figure. (see the revised manuscript). 

8) Line 166: I think you mean West Africa and not East Africa? 

Author’s Response:- Yes that was it was a writing error. Thank you very much for the correction. We accept your comment and corrected it accordingly(see the revised manuscript).

9) Table 2 shows the same number of women and the same number of children. If all children born to women in the last 5 years are included in the analysis, I would expect there to be more children included in the analysis than women included. Please clarify the discrepancy and add the number of women and number of children included in the analysis separately in the text (the flow chart in suggestion 2 would also help this point). 

Author’s Response:- Thank you very much for your concern related to the number of children and number of women included in this study. If a woman had more than one child five years preceding the survey, the most recent birth was taken because the number of children and women included in this study were the same.

10) Sample sizes from table 1 don’t match those listed in figure1 

Author’s Response:- Thank you very much for your comment. We accept your comment and corrected it accordingly(see the revised manuscript). 

11) The variable of the husband’s education is misrepresented. 31904 women who are not married should have a husband’s education as missing. 

Author’s Response:- Thank you very much for your comment. We accept your comment and corrected it accordingly( see the revised manuscript)

12) Please clarify the meaning of “No” for education status. Do you mean less than primary education? 

Author’s response:-No education means respondents who had no formal education (can not read and write)

13) Similar to 6: Please check variable for preceding birth interval. How is this defined for a woman’s first birth? If they are in the ≥24 months, this could be the reason for the unexpected results. 

Author’s response:- This variable very important variable for LBW. Before we fit the model, we exclude women who had one child. as we know stata uses complete case analysis during model fitting. Therefore this issue was resolved before any statistical analysis.

14) Table 4: Please present column % instead of row % for comparison between normal and low birthweight. 

Author’s response:-Thank you very much for your comment. We accept your comment and corrected it accordingly(see the revised manuscript).

Discussion

15) Overall the discussion could be improved to put the results in context for the reader. Consider restricting the discussion to focus on some key results. The conclusion discusses modifiable risk factors. One approach could be to discuss the modifiable risk factors and the context of interventions for these risk factors. Brining in the discussion of preterm and fetal growth restriction may also bring context to these results. 

Author’s response:-Thank you very much for your comment. We accept your comment and corrected it accordingly(see the revised manuscript).

16) All statements of explanation for these findings should be based on the literature and so need to be cited (line 227, 233, 237, 240, 249, 252, 254. 258). 

Author’s Response:- Thank you very much for your comment. We accept your comment and corrected it accordingly(see the revised manuscript).

17) Line 236: LBW associated with multiples is well established and it is an oversight here not to mention that multiples are almost always born early, and so increased risk for LBW for that reason. 

Author’s Response:- Thank you very much for your comment. We accept your comment and corrected it accordingly(see the revised manuscript).

Minor: 

In response to reviewer 3’s suggestions, remove the word determinants in line 38, 184

Introduction: 

Author’s response:-

1) Line 56: Please define LBW abbreviation – the first sentence could read, Low birth weight (LBW) is one of the…. 

Author’s response:- Author’s response:- Thank you very much for your comment. We accept your comment and corrected it accordingly(see the revised manuscript).

2) Line 61-63: The last sentence in the first paragraph doesn’t seem relevant to this overall summary that is at the SSA level. I suggest deleting or replacing high and low estimates for the region.

Author’s response:- Author’s response:- Thank you very much for your comment. We accept your comment and corrected it accordingly(see the revised manuscript).

3) Line 71 “well facts” should be reworded. Could say are well established. 

4) Line 78 Nutritional shouldn’t be capitalized.

5) Please provide citations for other LBW studies, meta-analyses systematic reviews referenced in line 83. 

6) 89-92. I am not convinced region-level estimates are very informative for interventions and policy decisions. 

Author’s Response:- Thank you very much for your comment. We accept your comment(3-6) and corrected it accordingly(see the revised manuscript).

Methods: 

7) Line 96-101: Please include the rationale for why these countries were selected. The list does not include the WHO African region or UN Africa region. 

Author’s response:-WHO had five Africa regions that are (North, West, East, Central, and South). We used the four African regions that represent sub-Saharan Africa (West, East, Central, and South). I hope the data flow char may answer this question(see the revised manuscript). 

8) Line 111: the specific coded values is not necessary to include. This line could just say “LWB was defined as a birth weight <2.5kg, and those ≥2.5kg were considered normal and above normal birthweights.” – this suggestion could be applied throughout the manuscript. 

Author’s response:-Thank you very much for your comment. We accept your comment(3-6) and corrected it accordingly(see the revised manuscript).

9) Line 122: “Based on kinds of literature” needs to be edited. It could be revised to say, potential risk factors for LBW were included based on a review of the literature. (Please include a citation). 

Author’s response:-Thank you very much for your comment. We accept your comment and corrected it accordingly(see the revised manuscript).

10) Line 156: If the P-value is used to base the decision on inclusion into the multivariable model, please include P-values with the bi-variate results. Because the study is overpowered for these detections, consider lowering the threshold of what is included in the multivariable analysis. If there were variables considered that did not meet this threshold, they should also be included in the manuscript. 

Author’s response:-Thank you very much for your comment. As we know the sample size is very large and we try to select variables based on literature. As commented previously, this analysis may had a problem of overfitting. We included this as a limitation. We believe that the confidence interval is better than the p-value since it measures precision. 

Results: 

1) Line 172: Please reword the description of highest low birth weight and lowest birthweight to make it easier to read. Consider saying highest LBW prevalence, or the lowest rate of LBW births. 

Author’s response:-Thank you very much for your comment. We accept your comment and corrected it accordingly(see the revised manuscript).

2) Line 193: Careful saying mothers were less likely to low birthweight. Consider revising to say “were less likely to deliver LBW infants than mothers without formal education: 

Author’s response:-Thank you very much for your comment. We accept your comment and corrected it accordingly(see the revised manuscript).

3) Line 201: Type on the confidence interval. Should be 0.91 instead of 0.81 (according to the table) 

4) Line 210: Typo Bing should be being. 

5) Table 2: please define media exposure. 

6) Table 4 CORs are missing on the last page for parity, sex of the child, type of births, iron supplementation. 

Authors’ response:-Thank you very much for your comment. We accept your comment from 3-6 and corrected it accordingly(see the revised manuscript). 

Discussion

7) Line 229: The statement about Brazil seems irrelevant for a paper focused on SSA- consider removing. The authors could consider comparing to other regions instead. 

Authors’ response:-Thank you very much for your comment. We accept your comment from 3-6 and corrected it accordingly(see the revised manuscript). 

8) consider the term multiples throughout instead of multiple pregnancies. 

Authors’ response:-Thank you very much for your comment. We accept your comment from 3-6 and corrected it accordingly(see the revised manuscript). 

Considerations 

1) It would be helpful to re-orient the risk factors so they are all risk factors or protective factors to help readability. 

Author’s response:-Thank you very much for your comment. There is only one/two variables change that is a risk almost all are in the protective size. We put the original presentation as it is.

2) Consider looking for trends in birth weight by year, may want to control for the year of the DHS survey. 

Author’s response:-Thank you very much for your comment. We only take a single survey for each country the recent one. There to do trend in needs more than one survey for each country that’s not our objective. 

3) Consider combining Table 2 and 4. 

Authors’ response:-Thank you very much for your comment. We accept your comment from 3-6 and corrected it accordingly(see the revised manuscript).

---

## [Decision Letter · Decision Letter 2]

24 Feb 2021

PONE-D-20-27726R2

Prevalence of low birth weight and its associated factor at birth in Sub-Saharan Africa: A Generalized Linear Mixed   Model

PLOS ONE

Dear Dr. Tadesse,

Thank you for submitting your manuscript to PLOS ONE. After careful consideration, we feel that it has merit but does not fully meet PLOS ONE’s publication criteria as it currently stands. Therefore, we invite you to submit a revised version of the manuscript that addresses the points raised during the review process.

I believe that the revisions made so far have improved the manuscript. Sorry, once again, to ask for further revision but the reviewer feels that this is important and I agree with them. The required revisions should not be too onerous and I look forward to seeing the newly revised version of the manuscript.

We look forward to receiving your revised manuscript.

Kind regards,

Clive J Petry, PhD

Academic Editor

PLOS ONE

Journal Requirements:

Reviewers' comments:

Reviewer's Responses to Questions

**Comments to the Author**

1. If the authors have adequately addressed your comments raised in a previous round of review and you feel that this manuscript is now acceptable for publication, you may indicate that here to bypass the “Comments to the Author” section, enter your conflict of interest statement in the “Confidential to Editor” section, and submit your "Accept" recommendation.

Reviewer #2: (No Response)

2. Is the manuscript technically sound, and do the data support the conclusions?

Reviewer #2: Yes

3. Has the statistical analysis been performed appropriately and rigorously? 

Reviewer #2: Yes

4. Have the authors made all data underlying the findings in their manuscript fully available?

Reviewer #2: Yes

5. Is the manuscript presented in an intelligible fashion and written in standard English?

Reviewer #2: Yes

6. Review Comments to the Author

Reviewer #2: Thank you to the authors for addressing the previous comments in such detail.

Critical

13) thank you for the explanation about the preceding birth interval variable. The sum of the birth interval variable equals the total sample size (<24=24,427, ≥24=178,450). If you dropped all primigravida women, that means the entire sample size is among multiparous women and this should be clarified in the methods, results, and discussion. If that is not correct, as I suspect the primigravida women are included in one of birth interval categories by mistake. Stata treats a missing variable as a very large number, so depending on how this categorical variable was created those with only one child may be included in the ≥24 category. You may want to include these women in their own category so they can be included in the analysis. Please clarify if this population is among multiparous women or correct the birth interval variable to include primigravida women in their own category.

Minor:

#12) Thank you for the clarification of No for education. I think it would be helpful to include this in the manuscript for readers that may have the same question.

7. PLOS authors have the option to publish the peer review history of their article (what does this mean?). If published, this will include your full peer review and any attached files.

Reviewer #2: **Yes: **Emily Deichsel

---

## [Author Response · Author response to Decision Letter 2]

25 Feb 2021

PLOS ONE 

Point by point response for editors/reviewers comments 

The manuscript title “Prevalence of low birth weight and its associated factor at birth in Sub-Saharan Africa: A Generalized Linear Mixed Model”

Manuscript ID: PONE-D-20-27726R3

Dear editor/reviewer. 

Dear all,

I would like to thank you for these fair, constructive, building, and improvable comments on this manuscript that would improve the substance and content of the manuscript. We considered each comment and clarification questions of reviewers on the manuscript thoroughly. My point-by-point responses for each comment and question are described in detail on the following pages. Further, the details of changes were shown by track changes in the supplementary document attached.

Response to Reviewers comments 

Reviewer #1

Critical

13) thank you for the explanation about the preceding birth interval variable. The sum of the birth interval variable equals the total sample size (<24=24,427, ≥24=178,450). If you dropped all primigravida women, that means the entire sample size is among multiparous women and this should be clarified in the methods, results, and discussion. If that is not correct, as I suspect the primigravida women are included in one of birth interval categories by mistake. Stata treats a missing variable as a very large number, so depending on how this categorical variable was created those with only one child may be included in the ≥24 category. You may want to include these women in their own category so they can be included in the analysis. Please clarify if this population is among multiparous women or correct the birth interval variable to include primigravida women in their own category.

Author’s Response:- Thank you very much for your condtractive and critical issues you raised. We accept you comments corrected accordingly. The reality is that this variables are included in the analysis and we try to explore our data and analysis and we included in the analysis at this stage as separate category. we included the primigravida in the analysis and we corrected in the table (table 3). See the revised manuscript. From a total of 202,878 samples 52,207 of them were primigravida and we include them in the analysis and we recoded this this variable 1-2, 3-5, and 6+ ( see revised table 3 and this number of sample size was edual to birth interval aand method session line 126) (<24=24,427, ≥24=178,450). The following stata output is the sample analysis when explore our data again. Many thanks your careful review of our manuscript. Therefore those population was part of our analysis and we included in the revised session. 

Minor:

#12) Thank you for the clarification of No for education. I think it would be helpful to include this in the manuscript for readers that may have the same question.

Author’s Response:- Thank you very much for comment and we accept your comment and included this concern in the revised session of our manuscript(see the revised manuscript line number 122-123 ).

---

## [Editor Report · Decision Letter 3]

26 Feb 2021

Prevalence of low birth weight and its associated factor at birth in Sub-Saharan Africa: A Generalized Linear Mixed   Model

PONE-D-20-27726R3

Dear Dr. Tadesse,

We’re pleased to inform you that your manuscript has been judged scientifically suitable for publication and will be formally accepted for publication once it meets all outstanding technical requirements.

Kind regards,

Clive J Petry, PhD

Academic Editor

PLOS ONE
---

## [Editor Report · Acceptance letter]

2 Mar 2021

PONE-D-20-27726R3 

Prevalence of low birth weight and its associated factor at birth in Sub-Saharan Africa: A Generalized Linear Mixed   Model 

Dear Dr. Tessema:

I'm pleased to inform you that your manuscript has been deemed suitable for publication in PLOS ONE. Congratulations! Your manuscript is now with our production department. 

Kind regards, 

on behalf of

Dr. Clive J Petry 

Academic Editor

PLOS ONE